# Poly (glycerol adipate) (PGA), an Enzymatically Synthesized Functionalizable Polyester and Versatile Drug Delivery Carrier: A Literature Update

**DOI:** 10.3390/polym11101561

**Published:** 2019-09-25

**Authors:** Sadie M.E. Swainson, Ioanna D. Styliari, Vincenzo Taresco, Martin C. Garnett

**Affiliations:** 1School of Pharmacy, University of Nottingham, University Park, Nottingham NG7 2RD, UK; 2School of Life and Medical Sciences, University of Hertfordshire, College Lane, Hatfield AL10 9AB, UK; i.d.styliari@herts.ac.uk; 3School of Chemistry, University of Nottingham, University Park, Nottingham NG7 2RD, UK

**Keywords:** poly(glycerol adipate), lipase, biodegradable, polymeric platforms, functionalization

## Abstract

The enzymatically synthesized poly (glycerol adipate) (PGA) has demonstrated all the desirable key properties required from a performing biomaterial to be considered a versatile “polymeric-tool” in the broad field of drug delivery. The step-growth polymerization pathway catalyzed by lipase generates a highly functionalizable platform while avoiding tedious steps of protection and deprotection. Synthesis requires only minor purification steps and uses cheap and readily available reagents. The final polymeric material is biodegradable, biocompatible and intrinsically amphiphilic, with a good propensity to self-assemble into nanoparticles (NPs). The free hydroxyl group lends itself to a variety of chemical derivatizations via simple reaction pathways which alter its physico-chemical properties with a possibility to generate an endless number of possible active macromolecules. The present work aims to summarize the available literature about PGA synthesis, architecture alterations, chemical modifications and its application in drug and gene delivery as a versatile carrier. Following on from this, the evolution of the concept of enzymatically-degradable PGA-drug conjugation has been explored, reporting recent examples in the literature.

## 1. Introduction

Interest in the use of enzymes in polymer synthesis as a greener alternative to traditional chemical polymerization has been growing for a number of years [1]. There are many advantages associated with enzymatic synthesis, including but not restricted to: (a) mild reaction conditions, (b) catalysts with low toxicity, high and tunable activity which are often recyclable, (c) the avoidance of toxic heavy metal catalysts, (d) often good linearity of products due to steric hindrance at the enzyme active site, (e) few by-products, and (f) less need for protection and deprotection steps [2,3,4,5].

The enzymatic synthesis of polyesters [6], and in particular poly (glycerol adipate) (PGA), has resulted in an elegant, easy and versatile strategy to synthesize a functionalizable, biocompatible (in *vitro* and in *vivo*) and biodegradable class of polymers [7,8]. PGA shows a precise amphiphilic balance within the repetitive unit, allowing self-assembly into NPs in water by simple nanoprecipitation, without the use of additional stabilizers. The chemo- and regioselectivity of lipase allows the hydroxyl moiety of the PGA backbone to remain intact, without the need for tedious and complicated protection and deprotection steps [9]. This hydroxyl moiety results in a polymer open to a variety of further functionalizations via simple and accessible chemistry. Based on the promising properties of PGA for drug delivery, a number of research groups have focused on establishing strategies for controlling the polymeric function and adding modifications to enhance material properties. This work has led to a series of literature precedents/guidelines for future researchers interested in exploiting the advantages of PGA, and these will be summarized briefly here.

## 2. Synthesis of PGA

Following a solvent free pathway, it has been demonstrated that, using divinyl adipate (DVA) and glycerol, by tuning the reaction temperature it is possible to produce either highly linear or densely branched PGA [10,11,12,13]. However, number-average molecular weights (*M*n) lower than 5000 Da (<25 units) were observed even after over 24 h of reaction with lipase. Novozym 435 (Sigma Aldrich), an acrylic lipase resin from *Candida antarctica*, was used for all synthesis. Kallinteri et al. had the perception to dilute the reaction mixture with THF, with the aim to reduce the viscosity of the system [14]. They also moved from a magnetic to a mechanical stirrer, in order to increase the size of the reaction pot with a view to scale-up the synthesis and thus facilitate further functionalization [14]. In this work, Kallinteri et al. demonstrated that higher polymer molecular weight has a remarkable effect on the encapsulation efficiency and drug loading of the nano-formulation.

Following these alterations, a range of molecular weights could be explored by simply changing the reaction time. Based on these initial screenings, the Kressler and Garnett groups have been improving reaction conditions and gaining an understanding of the effects of the reaction variables on the final properties of PGA [15,16]. In particular, Taresco et al., through an extensive screening investigation, established how the temperature of the reaction affects the final architecture of the polymer; at 40 °C there was <5% of branching while at 70 °C; >30% of branching was detected [17]. They also evaluated how this composition can alter thermal properties, such as the glass transition temperature (*T*g), and physical-chemical properties of the polymeric backbone, such as water contact angle and polymer molecular weight (Figure 1).

Naoluo et al. [18] suggested dimethyl adipate (DMA) may be more appropriate than DVA for scaling up the synthesis of PGA, as DMA is cheap and readily available. DMA shifts the equilibrium of the reaction towards the polymer; methanol is produced as a by-product and cannot be easily removed. A complex experimental set up with molecular sieves in a soxhlet apparatus was employed to remove the methanol during the reaction. The resultant polymer was found to have a lower molecular weight than PGA produced with DVA, despite a longer reaction time being employed. Additionally, Korupp et al. [12] and Iglesias et al. [19] produced PGA using adipic acid as a starting material; as with DMA, the molecular weight of the PGA produced was significantly lower than that seen by the Garnett group when producing PGA with DVA [14,17]. Korupp et al. also demonstrated that optimizing reaction conditions, such as temperature, stirrer rate and reaction time, allows synthesis of PGA batches of up to 500 g, with ~95% monomer conversion [12].

## 3. Modifications of PGA

An interesting chemical variation of the PGA backbone during the contact period of the reaction mixture with lipase has been introduced by the Hutcheon and Saleem groups. In one reaction pot, they performed the usual enzymatic polycondensation between DVA and glycerol with an enzymatic ring opening polymerization (ROP), adding ω-pentadecalactone (PDL) to the initial mixture [20,21]. The resultant PGA-co-PDL polymer showed different physical properties to the bare PGA, mainly in terms of thermal properties and amphiphilicity of the copolymer. To further modify the hydrophobic/hydrophilic balance of the glycerolated polymer, PEG chains have been introduced in the reaction mixture [22]. PGA-co-PDL and the PEGylated variation have shown good propensity to be formulated into nanoparticles (NPs) or microparticles (MPs). These biodegradable particles have been investigated for the encapsulation or adsorption of small active molecules, such as indomethacin, ibuprofen and sodium diclofenac, and macromolecules such as proteins [23,24,25,26,27].

Jbeily et al. converted the PGA into an atom transfer radical polymerization (ATRP) macroinitiator, and subsequently glycerol mono-methacrylate was polymerized by ATRP from the PGA backbone, yielding an amphiphilic graft copolymer able to self-assemble in water [28]. The free hydroxyl pendant groups have also been employed as ROP initiators for the production of polycaprolactone-grafted-PGA copolymers with recrystallizable side chains [29]. The subsequent “click” coupling of PEGylated chains led to hybrid materials able to form worm-like nanoaggregates [30]. These synthetic tandem strategies provide a facile opportunity to synthesize well-defined amphiphilic graft hybrid copolymers, with the potential to simultaneously deliver hydrophilic and hydrophobic drugs [30].

The most pursued and most versatile strategy to tailor the characteristics of PGA for specific applications, i.e., to alter the physical properties of the polymer as well as enhance its chemical diversity and tune polymer (macro)molecule interactions, is the post-polymerization-functionalization. Kallinteri and Garnett et al. [14] not only improved the reaction conditions for the synthesis of the polymer backbone, but also had the intuition to move focus from the pure fundamental chemistry of the polymeric backbone towards the functionalization of the side group, in order to enhance the properties of the final material. Acylation of the backbone with 20–100% caprylic acid (C_8_) and stearic acid (C_18_) produced a hydrophobic environment for nanoparticle formation and drug incorporation. The acylation was shown to affect the particle size and encapsulation efficiency without significantly affecting the viability of HL–60 and HepG_2_ cells (Figure 2).

PGA acylation with fatty acids has been intensively investigated both in terms of physical properties, self-assembling in NPs and for delivery of lipophilic and hydrophilic drugs [31,32,33,34]. The variation of both the fatty acid nature (butyrate, octanoate, laurate, stearate, behenate and oleate) and degrees of substitution have been analyzed for their influence on the final polymer amphiphilic balance, affecting NP shape, drug interactions, NP metabolism and cell uptake [33,35,36]. Weiss et al. [33] reported that a low substitution of lauric, stearic or behenic acid decreased nanoparticle size relative to unmodified PGA, suggesting interactions of fatty acid chains in the particle core enhanced the packing of particles (Table 1). These particles were found to have non-spherical shapes with an internal lamellar-like structure. At higher substitutions particles tended to be ellipsoidal or spherical with an increased particle size, reflecting either an increase in the space taken up by the acyl groups or an increase in aggregation number (Figure 2). Interestingly, the chain length had no clear impact on the particle size.

The ability of bare PGA, and PGA with the polymer backbone substituted with varying amounts of pendant C_18_ (stearate) and C_8_ (octanoate) chain length acyl groups, to encapsulate the water soluble drug dexamethasone phosphate (DXMP) into NPs was investigated in two papers from the Garnett group [14,37]. In both papers, polymers modified with C_8_ were found to give the greatest improvement in encapsulation efficiency, with the level of substitution also proving to be a key factor (Figure 3). Puri et al. noted that the level of substitution affected the release of the DXMP, with a controlled release profile observed from particles made of PGA with a 100% C_8_ substitution [14,37]. An agreement with the experimental trend in encapsulation efficiency has been found for a series of acylated PGAs modelled computationally. In particular, while C_18_ chains promote favorable interactions with DXMP, having too many C_18_ chains decreases solubility, preventing the polymer from interacting with surrounding DXMP molecules. It was therefore clear that the hydrophilic/hydrophobic balance of the grafted copolymers was fundamental in the encapsulation efficiency and drug release [37,38].

To further remark on the versatility of the acylated PGA modification, Tchoryk et al. developed a revolutionary method to monitor NP penetration through 3D spheroid cell cultures [39], building on previous work in 3D models carried out by Meng et al. [40,41] Interestingly, NPs of around 100 nm, prepared from PGA–C_18_ end functionalized with a PEGylated chain, showed the same penetration as much smaller (50 nm) commercial model polystyrene NPs, suggesting the advantages of material flexibility shown by PGA. This flexibility effect, combined with the in *vivo* distribution of PGA NPs [7] and the exceptional ability of acyl-PGA alterations to encapsulate hydrophobic and hydrophilic drugs, opens up a wide range of possible applications in drug delivery [39].

## 4. Polymer Drug Conjugates

More recently, polymer-drug conjugates using PGA have been actively studied and designed as advanced drug delivery systems. The first example of this application was the conjugation of PGA–co–PDL to ibuprofen, where it was shown that most of the drug was attached to the polymer in a stable manner when incubated in physiological buffer [42]. PGA has also been directly coupled to indomethacin, producing a polymeric pro-drug composite able to self-assemble into NPs and to release the drug cargo in a sustained and controlled way, showing potential to improve therapy of inflammation and associated diseases [43,44].

By coupling PGA with a small number of aromatic N-acetyl amino acids, a novel class of biodegradable grafted polyesters with tunable physical properties have been developed, with the intention to widen the range of possible interactions with drugs and biological macromolecules [45]. This approach towards modifying the polymer was used by Suksiriworapong et al., who reported for the first time a polymer-anticancer drug conjugate based on PGA through the successful conjugation of methotrexate (MTX), avoiding the use of intermediate linkers. MTX–PGA conjugates were formed in a controlled manner, with various molar ratios of MTX [46]. The MTX–PGA conjugate self-assembled into nanoparticles with a size dependent on the amount of conjugated MTX and the pH of medium. NPs were chemically stable against hydrolysis at pH 7.4 over 30 days but were susceptible to enzymatic hydrolysis, thus selectively releasing free MTX. The 30% MTX-PGA nanoparticles exhibited only slightly less potency than free MTX in 791T cells. This was a significant improvement compared to previous reports of human serum albumin-MTX conjugates which were only one three hundredth the potency of free MTX. In addition, the MTX nanoparticles showed 7 times higher toxicity to Saos-2 cells than MTX in 2D and 3D cell experiments (Scheme 1) [46]. These easily fabricated PGA–MTX conjugates, where the enzymatic degradability has replaced the need for a linker group, may become an effective new strategy for the development of polymeric pro-drug formulations.

## 5. Recent Experiments with PGA from Literature

More recently, PGA was used to evaluate the validity of commonly used solubility parameters [47]. The choice of PGA for this experimental work was due to the wide range of physicochemical properties which can be obtained by simple polymer modification. Both experimental and in silico methods were employed in parallel using a large varied data set that looked into the miscibility of a range of drugs with different polymers, amongst which was PGA, PGA–C_4_, PGA–C_8_ and amino acid-modified PGA–Phe. Due to the high number of possible combinations, a miniaturized high throughput screening was performed using a 2D inkjet assay [48] to assess the miscibility limits of the polymer-drug mixtures. The in-silico study using ab initio methods was unable to predict the drug-polymer compatibility, demonstrating the need for new computational methods. PGA in turn provides a large library of polymers for possible use in the high throughput screening developed in this paper to select the most appropriate polymer for further studies in a range of different applications. In addition, PGA and its modifications have shown good propensity to produce printable polymeric ink, even at relatively high concentration and in different printing scenarios: both in a solid state and for nanosuspension screening [49].

It has often been suggested that PGA will undergo enzymatic degradation as a result of the presence of an ester bond in the polymer backbone and the enzymatic synthesis used to produce the polymer which uses the reverse reaction of a degradative enzyme. Recently, the degradation of PGA in a range of enzymes has been extensively characterized by Swainson et al. [50] (Scheme 2). Nanoparticles produced with unmodified PGA and PGA modified with amino acids, carboxyfluorescein and PEG were exposed to six enzymes and the change in size monitored over time by dynamic light scattering. A clear difference in the susceptibility of these particles to enzymatic degradation was observed. Following this, the release of a conjugated fluorescent dye, carboxyfluorescein, was seen to increase in the presence of lipase and elastase. Finally, the structural changes to the polymer were studied using fourier transformed infra-red spectroscopy (FTIR), gel permeation chromatography (GPC) and nuclear magnetic resonance (NMR), to gain an understanding of the way PGA breaks down in the presence of lipase, elastase and esterase. The degradation behavior evidenced in this work suggests the initial hypotheses about the degradation of PGA were correct and highlights the potential of this polymer, both modified and unmodified, as a drug delivery platform.

## 6. Clinical Outlook

As yet, there are no reports of PGA use in clinical trials. Taresco et al. reported that unmodified PGA was well tolerated in a chronic oral dosing study in rats, with the no observed adverse effect level reported to be 1000 mg/kg/day [17]. It should be noted, however, that this data was not published, and there are no published examples of PGA applications tested in *vivo*. The previous in *vitro* studies in which PGA was shown to have no negative effect on cell viability [8,14], coupled with the promising results of studies such as that by Suksiriworapong et al. [46], suggest that progression towards in *vivo* studies, and ultimately clinical trials, can be expected in the near future; PGA shows great potential as a drug delivery platform, but as yet this potential is unrealized.

## 7. Conclusions

In the present manuscript we have extensively retraced the evolution of the enzymatic synthesis of PGA, discussing and summarizing the most relevant literature to shine a light on the underlying physical-chemical properties of PGA and its promising use as a carrier polymer in drug delivery. Critical chemical modifications making use of the PGA free hydroxyl side group to enhance the material properties have been discussed in this work. This is a key advantage of PGA compared to other readily available biodegradable polymers, as it has the potential to expand the range of drugs which can be easily encapsulated and retained within drug delivery systems. Furthermore, the ready biodegradability of PGA in comparison with other polymers, opens up new ways of using this polymer in drug delivery, e.g., in more effective polymer drug conjugates. Consideration of these factors results in a brief but detailed guideline for future research groups interested in using PGA in different fields. Taking into account all the relevant physicochemical and biological properties of NPs (or achievable devices) based on PGA and PGA chemical alterations, combined with the exceptional ability of PGA and its alterations to encapsulate/amorphously stabilize hydrophobic and hydrophilic drugs, it is clear that PGA has great potential for a wide range of possible applications in drug delivery and personalized medicine, many of which remain to be explored fully.

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
