# Peer review of "Poly (glycerol adipate) (PGA), an Enzymatically Synthesized Functionalizable Polyester and Versatile Drug Delivery Carrier: A Literature Update"

_polymers, 2019, doi:10.3390/polym11101561_

Round 1

Reviewer 1 Report

Swainson et al wrote a short review on the enzymatic synthesis of polyesters as possible carriers for drud delivery.

The subject is potentially interesting for the readers of polymers but the manuscript is too superficial and not well organized. I suggest to reorganize the manuscript according with the following points:

1. First of all there are only two paragraphs: a long introduction (par. 1) and short conclusions (par.4!). I suggest that par. 1 is divided in different sections like: 2. enzymatic synthesis, 3. self assembly or nanoparticle formation, 4. drug delivery. These are just an example, but would help the reader to better follow the logic sequence of the review.

2. I also suggest to go in some more details, as for example list all the different source of lipases used for the polymerization, or more details on the self-assembly of the polymers, etc.

3. English needs to be improved in some parts.

4. there are some minor points, as abbreviations to be corrected: Molecular weigth cannot be "Mn" but "MW". Similarly Glycerol cannot be abbreviated "GA" or thermal "Tg".

Author Response

Reviewer 1

Swainson et al wrote a short review on the enzymatic synthesis of polyesters as possible carriers for drud delivery.

The subject is potentially interesting for the readers of polymers but the manuscript is too superficial and not well organized. I suggest to reorganize the manuscript according with the following points:

First of all there are only two paragraphs: a long introduction (par. 1) and short conclusions (par.4!). I suggest that par. 1 is divided in different sections like: 2. enzymatic synthesis, 3. self assembly or nanoparticle formation, 4. drug delivery. These are just an example, but would help the reader to better follow the logic sequence of the review. I also suggest to go in some more details, as for example list all the different source of lipases used for the polymerization, or more details on the self-assembly of the polymers, etc. English needs to be improved in some parts. there are some minor points, as abbreviations to be corrected: Molecular weigth cannot be "Mn" but "MW". Similarly Glycerol cannot be abbreviated "GA" or thermal "Tg".

We agree with the suggestion of reviewer 1 and have added sections to the review accordingly. Additionally, several areas of the paper have been reorganised to give a more logical progression through the subject matter. The source of lipase tends to be the same for all PGA synthesis and so has been included to provide more detailed information. The review has been carefully proof read to ensure the writing style is correct and consistent. The abbreviation Mn has been more clearly defined; it was used here to represent the number-average molecular weight, obtained from GPC measurements. The abbreviation GA has been removed and replaced with glycerol in all instances. Finally, the sentence in which Tg was used has been re-written to clarify that it is the glass transition temperature which can be affected by synthesis conditions.

Reviewer 2 Report

This is an interesting review of the recent literature and could be published as it is. However I have found some small errors, which should be corrected before publication. These are as follows:

1./ Authors do use big and small letters when naming reagents and enzymes. It is better to use small letters, but anyway the naming should be standarized;

2./ formulas of branched polymers inserted to Scheme 1 are simply ugly;

3./ line 166 should be4 ibuprofen where;

4./ line 199 should be disscussion on the need(?);

5./ line 233 should be results in.

Author Response

This is an interesting review of the recent literature and could be published as it is. However I have found some small errors, which should be corrected before publication. These are as follows:

1./ Authors do use big and small letters when naming reagents and enzymes. It is better to use small letters, but anyway the naming should be standarized;

2./ formulas of branched polymers inserted to Scheme 1 are simply ugly;

3./ line 166 should be4 ibuprofen where;

4./ line 199 should be discussion on the need(?);

5./ line 233 should be results in.

Thank you for your comments, we have made corrections as suggested. The naming of reagent and enzymes has been standardised, with lower case used throughout. Scheme 1 has been replaced with an alternative figure from the same paper. The small typographical corrections have either been made as suggested or the sentence re-written (point 5) for extra clarity.

Reviewer 3 Report

The review is well written. Most of the aspects regarding PGA synthesis is well captured in the review. However, the impact of the review can be further enhanced if the authors can include clinical translation aspects. Like scale-up synthesis stratergies, GMP for PGA. Additionally authors can provide a tabular column like list of products avaliable in the market or uncer clinical trials for healthcare.

Author Response

The review is well written. Most of the aspects regarding PGA synthesis is well captured in the review. However, the impact of the review can be further enhanced if the authors can include clinical translation aspects. Like scale-up synthesis stratergies, GMP for PGA. Additionally authors can provide a tabular column like list of products avaliable in the market or uncer clinical trials for healthcare.

Thank you for the suggestion. A section on the clinical outlook for PGA has been added at the end of the review. To the best of our knowledge, there are no published clinical trials for PGA, however, we have discussed a chronic oral dosing study carried out in rats. Information about the scale up of PGA has also been added to the synthesis section for added clarity.

Round 2

Reviewer 1 Report

The authors have improved the quality of their manuscript. I only have a minor concern at pag 2 lines 52-53 there is an error. Here is the correction:

Novozym 435 is a lipase from Candida antarCtica B immobilized on an acrylic resin.

Author Response

We have now changed the single letter typo as noted by the reviewer.
